



# Hierarchical Sensitivity Analysis for Large Scale Process-based Hydrological Modeling with Application in an Amazonian Watershed

Haifan Liu[1], Heng Dai[2*], Jie Niu[2*], Bill X. Hu[2], Han Qiu[3], Dongwei Gui[4], Ming Ye[5], Xingyuan Chen[6], Chuanhao Wu[2], Jin Zhang[2], and William Riley[7]

[1]School of Water Resources and Environment, China University of Geosciences, Beijing, 100083, China.
[2]Institute of Groundwater and Earth Sciences, Jinan University, Guangzhou 510632, China.
[3]State Key Laboratory of Desert and Oasis Ecology, Xinjiang Institute of Ecology and Geography, Chinese Academy of Sciences, Urumqi 830011, China
[4]Department of Civil and Environmental Engineering, Michigan State University, East Lansing, MI. USA.
[5]Department of Earth, Ocean, and Atmospheric Science, Florida State University, Tallahassee, FL 32306, USA.
[6]Pacific Northwest National Laboratory, Richland, WA 99352, USA.
[7]Earth Sciences Division, Lawrence Berkeley National Laboratory, Berkeley, California, USA.

*Correspondence to*: Heng Dai (heng.dai@jnu.edu.cn) and Jie Niu (jniu@jnu.edu.cn)

**Abstract.** Sensitivity analysis is an effective tool for identifying important uncertainty sources and improving model calibration and predictions, especially for integrated systems with heterogeneous parameter inputs and complex processes coevolution. In this work, an advanced hierarchical global sensitivity analysis framework, which integrates the concept of variance-based global sensitivity analysis with hierarchical uncertainty framework, was implemented to quantitatively analyze several uncertainties of a three-dimensional, process-based hydrologic model (PAWS). The uncertainty sources considered include model parameters, model structures (with/without overland flow module), and climate forcing. We apply the approach in a ~9000 km$^2$ Amazon catchment modeled at 1 km resolution to provide a demonstration of multiple uncertainty source quantification using a large-scale process-based hydrologic model. The sensitivity indices are assessed based on three important hydrologic outputs: evapotranspiration ($ET$), ground evaporation ($E_G$), and groundwater contribution to streamflow ($Q_G$). It is found that, in general, model parameters (especially those within the streamside model grid cells) are the most important uncertainty contributor for all sensitivity indices. In addition, the overland flow module significantly contributes to model predictive uncertainty. These results can assist model calibration and provide modelers a better understanding of the general sources of uncertainty in predictions of complex hydrological systems in Amazonia. We demonstrated a pilot example for comprehensive global sensitivity analysis of large-scale complex hydrological and environmental models in this research. The hierarchical sensitivity analysis methodology used is mathematically rigorous and capable of being implemented into a variety of large-scale hydrological models with various sources of uncertainty.





## 1 Introduction

Large-scale three-dimensional Process-Based Hydrologic Models (PBHMs) are being increasingly applied not only for evaluating watershed hydrologic responses to climate forcing, but also for understanding ecosystem energy balance, biogeochemistry, and ecological functioning from basin to continental scales (Shen et al., 2013; Maxwell et al., 2014; Riley

and Shen 2014). PBHMs can simulate interacting states and fluxes of the integrated surface and subsurface hydrological processes and accurately predict large-scale complex hydrologic system behaviours (Niu et al., 2014; Shen et al., 2014; Niu and Phanikumar, 2015; Niu et al., 2017). Distributed flow pathways, e.g., evapotranspiration (*ET*), overland flow, channel runoff, etc., can be distinguished by PBHMs simulations (Beven, 2002). In addition, the governing equations for subsurface flow are explicitly solved in PBHMs; thus, they can simulate detailed hydrological processes including root extraction,

infiltration, soil evaporation, and groundwater discharge and recharge in the vadose zone (Maxwell et al., 2014). However, complexities and uncertainties inherent in PBHMs structures, heterogeneous model parameters, heterogeneous data sources (e.g., elevation, soil properties, groundwater conductivities), and climate forcing may produce high uncertainties in the modeling outputs (Shen et al., 2014; Qiu et al., 2019).

Uncertainty in numerical modeling is inevitable and important (Neuman, 2003; Rojas et al., 2010), especially for PBHMs

that represent a high level of physical process complexity. Previous comprehensive sensitivity analyses of large watershed models rarely apply complex PBHMs (Emery et al., 2016). Uncertainties could emerge from multiple sources, including climate forcing, lack of data or knowledge to capture some characteristics of the system, heterogeneity in natural characteristics, and equifinality of conceptual model parameters, etc. (Beven, 1993; Beven and Freer, 2001; Nossent et al., 2011; Refsgaard et al., 2007, 2012; Ye et al., 2008; Sulis et al., 2011). Sensitivity analysis becomes important to identify the

most influential uncertainty sources so that limited resources can be used to maximumly reduce model predictive uncertainty (Neuman, 2003; Saltelli et al., 2010; Lu et al., 2012; Song et al., 2015).

In general, sensitivity analysis can be separated into local and global methods. The main limitation of local sensitivity analysis is that its results are only valid for a small range of parameter values (Gedeon et al., 2012; King and Perera, 2013; Wainwright et al., 2014; Dai and Ye, 2015). Comparing to the local method, global sensitivity analysis can provide

sensitivity estimates for the entire range of uncertain parameters values (Saltelli et al., 2000, 2010; Razavi and Gupta 2015, 2016). Because of this advantage, global sensitivity analysis has gained popularity in recent modeling analyses despite its high computational cost (Hamby, 1994; van Griensven et al., 2006; Sulis et al., 2011; Baroni et al., 2014). Among different global sensitivity analysis methods, the variance-based method has been widely used because its ability to accurately quantify the importance of uncertain parameters while considering their interactions (Saltelli and Sobol, 1995; Zhang et al.,

2013; Dai and Ye, 2015).

There is still a lack of research for quantitative and representative global sensitivity analysis using large-scale PBHMs. Most sensitivity analysis research for hydrological models has used local sensitivity analysis methods (Chavarri et al., 2013; de



Paiva et al., 2013; Nijssen et al., 2001). Moreover, the conventional global sensitivity analysis only includes the uncertainty from model parameters and ignores other important sources of hydrological model uncertainties, including scenario uncertainty (caused by alternative unpredictable future climate conditions, e.g., precipitation, radiation intensity, temperature) and structural uncertainty (caused by different interpretations of the real situation and reflecting on the plausible models (Ye et al., 2005; Makler-Pick et al., 2011; Neumann, 2012; Dai and Ye, 2015; Song et al., 2015; Dai et al., 2017a,b )).

In this research, we applied a new hierarchical global sensitivity analysis approach to a PBHM (the Process-Based Adaptive Watershed Simulator; PAWS) at a watershed in the Amazon and considering multiple uncertainty sources. Hierarchical sensitivity analysis was first proposed by Dai and Ye (2015) and then applied to a groundwater modeling analysis for the Hanford 300 area in Washington, U.S. (Dai et al., 2017a). This new methodology integrates the concept of variance-based global sensitivity analysis with the hierarchical uncertainty quantification framework (Draper et al., 1999; Dai and Ye, 2015) to quantify sensitivity of important processes to model parametric and structural uncertainty. This new sensitivity analysis method considers three important sources of uncertainties (i.e., scenario, structural, and parametric) involved in hydrological models and dramatically decreases computational costs by combining uncertain inputs based on their characteristics and inter-dependencies. In this study, we revised the hierarchical sensitivity analysis method and defined a set of sensitivity indices to group and accurately quantify the importance of different uncertainty sources in PAWS when simulating hydrological processes in an Amazonian watershed.

PAWS has been applied extensively in many watersheds, e.g., the large watersheds in Michigan state, USA (Niu et al., 2014, 2017; Ji et al., 2015; Shen et al., 2013, 2014, 2016; Qiu et al., 2019) and the watershed in Amazon basin (Niu et al., 2017), and demonstrates high efficiency and good performances. We applied PAWS here in the Amazon because it consists of more than half of the tropical rainforests (Morley, 2000) and plays an essential role in the world carbon (Richey et al., 2002; Phillips et al., 2009; Lintner et al., 2017) and water (Fearnside, 2005; Phillips et al., 2009) cycles. Amazonian forest hydrology can affect many processes, including nutrient budgets (Lesack and Melack 1996), production and consumption of carbohydrates (Pegoraro et al., 2006), and root-zone moisture availability for plants (Oliveira et al., 2005; Tang et al., 2015). A large number of field (Lesack 1993; Leopoldo et al., 1995; Tomasella et al., 2008) and numerical modeling (Coe et al., 2008; Sorribas et al., 2016; Yamazaki et al., 2012) studies have been done to improve our understanding of hydrologic processes in the Amazon. As some studies indicated that groundwater is the key controller of the Amazon hydrology (Leopoldo et al., 1995; Hodnett et al., 1997a,b; Beighley et al., 2009; Pokhrel et al., 2014; Niu et al., 2017), some researchers have already implemented simulations containing subsurface water in Amazon (Miguez-Macho and Fan, 2012a,b). The severe hydrological hazards that frequently occurred in Amazon in the past, such as the droughts in 2005, and 2010 and the floods of 2009 (Tomasella et al., 2008; Marengo et al., 2010; Espinoza et al., 2011), also raise the requirement of including climate scenario uncertainty and related model uncertainty in sensitivity analyses for the Amazon region.

The PAWS model applied in this work was originally used in Niu et al. (2017) for simulating the hydrologic cycle in an Amazonian watershed. Here we build on that work by combing model system uncertainty sources into three groups: climate





scenarios (precipitation, radiation and temperature), model structure (with and without overland flow module), and parameters, including soil saturated hydraulic conductivity $K_s$ (m day$^{-1}$), unconfined aquifer hydraulic conductivity $K_1$ (m day$^{-1}$), confined aquifer hydraulic conductivity $K_2$ (m day$^{-1}$), Van Genuchten equation parameters $\alpha$ (unitless) and $N$ (unitless) (Genuchten, 1980). Because of the high complexity and dimensionality of this model, the highly efficient parameter

sampling method, Latin hypercube sampling (LHS) was applied to obtain parameter samples and estimate the new sensitivity indices. Through implementing the hierarchical sensitivity analysis method, we aim to provide a pilot example of comprehensive global sensitivity analysis for the large-scale PBHMs and investigate the most important uncertainty source for modeling hydrological processes in the Amazon.

In section 2, we describe the hierarchical sensitivity analysis method and Latin hypercube sampling. The introduction of the

study site and details about the implementation of sensitivity analysis using PAWS are exhibited in section 3. Then we display and discuss the results of sensitivity indices for three model outputs of interest: evapotranspiration ($ET$), ground evaporation ($E_G$), and groundwater contribution to stream ($Q_G$) in section 4. Section 5 is the summarization of the key findings of this research.

## 2 Methodology

We start with the introduction of hierarchical uncertainty framework and the new sensitivity indices of different uncertainty sources in Section 2.1. Thereafter the numerical evaluation of new sensitivity indices using Latin hypercube sampling is exhibited in Section 2.2.

### 2.1 Hierarchical uncertainty framework and new sensitivity index

In this study, we built the hierarchical uncertainty framework containing three layers (Fig. 1) to represent the uncertain

inputs and outputs of the PAWS model system used in this research. Fig. 1 demonstrates three major uncertain inputs for generating the model output are placed in appropriate positions: (1) the climate scenarios, (2) the conceptualization of the numerical model, and (3) model parameters. The climate scenarios (e.g., precipitation, radiation intensity, temperature) are the driving forces of the model system. Numerical model uncertainty represents the different possible conceptualization describing the study site. Each model can contain a different set of parameters and their values are subjected to parameter

uncertainty (Meyer et al., 2007). These three sources of uncertainty were put into the proper level of hierarchy in the uncertainty framework based on their dependence relationships.

For a model: $\Delta = f(X) = f(X_1, ..., X_m)$, where $\Delta$ is the model output and $X = \{X_1, ..., X_m\}$ is a group of model uncertain inputs, using the law of total variance, the total variance of $\Delta$ can be decomposed as:

$$V(\Delta) = V_{X_i}\left(E_{\mathbf{X}_{-i}}\left(\Delta | X_i\right)\right) + E_{X_i}\left(V_{\mathbf{X}_{-i}}\left(\Delta | X_i\right)\right), \tag{1}$$





where the first term of partial variance on the right-hand side is the within-$X_i$ partial variance and represents the variance

contribution by $X_i$, and $\mathbf{X}_{\sim i}$, represents all the inputs except $X_i$. The second term on the right-hand side represents the

variance contributed by the model inputs excluding $X_i$ as well as the interactions of all the inputs. Based on the definition of

first-order sensitivity index $S_i = \dfrac{V_{X_i}(E_{\mathbf{X}_{\sim i}}(\Delta \mid X_i))}{V(\Delta)}$ , the percentage of uncertainty contributed by input $X_i$ can be quantified

accurately.

Using the hierarchical framework in Fig. 1, the variance-based sensitivity analysis method enables the decomposition of the

total variance into individual contributors following:

$$
\begin{aligned}
V(\Delta) &= V_{\mathbf{CS}}\left(E_{\sim\mathbf{CS}|\mathbf{CS}}\left(\Delta|\mathbf{CS}\right)\right) + E_{\mathbf{CS}}\left(V_{\sim\mathbf{CS}|\mathbf{CS}}\left(\Delta|\mathbf{CS}\right)\right) \\
&= V_{\mathbf{CS}}\left(E_{\mathbf{NM,PR}|\mathbf{CS}}\left(\Delta|\mathbf{CS}\right)\right) + E_{\mathbf{CS}}\left(V_{\mathbf{NM,PR}|\mathbf{CS}}\left(\Delta|\mathbf{CS}\right)\right)
\end{aligned}
\tag{2}
$$

where $\mathbf{CS}$ represents the set of alternative climate scenarios, $\mathbf{NM}$ represents the multiple plausible numerical models, and

$\mathbf{PR}$ is the uncertain parameters. The term $\sim\mathbf{CS}$ represents the uncertain inputs excluding climate scenarios, which are

$\mathbf{NM}$ and $\mathbf{PR}$ in this study. The term $\mathbf{NM,PR}|\mathbf{CS}$ represents the change of combination of model and parameter under

one fixed climate scenario. The first term of partial variance on the right-hand side of this equation represents the variance

caused by multiple climate scenarios. The second term on the right-hand side is the partial variance caused by other

uncertain inputs and can be further decomposed as:

$$
V_{\mathbf{NM,PR}|\mathbf{CS}}\left(\Delta|\mathbf{CS}\right) = V_{\mathbf{NM}|\mathbf{CS}}\left(E_{\mathbf{PR}|\mathbf{NM,CS}}\left(\Delta|\mathbf{NM,CS}\right)\right) + E_{\mathbf{NM}|\mathbf{CS}}\left(V_{\mathbf{PR}|\mathbf{NM,CS}}\left(\Delta|\mathbf{NM,CS}\right)\right),
\tag{3}
$$

where the first partial variance term on the right-hand side of this equation represents the uncertainty contributed by multiple

plausible models. The subscripts $\mathbf{NM}|\mathbf{CS}$ and $\mathbf{PR}|\mathbf{NM,CS}$ refer to the change of models under one climate scenario and

the change of parameters under one model and climate scenario respectively. The second term represents the within-model

partial variance which is caused by the uncertain parameters. By substituting Eq. (3) back into Eq. (2), we can get that:

$$
\begin{aligned}
V(\Delta) &= E_{\mathbf{CS}}\left(E_{\mathbf{NM}|\mathbf{CS}}V_{\mathbf{PR}|\mathbf{NM,CS}}\left(\Delta \mid \mathbf{NM,CS}\right) + V_{\mathbf{NM}|\mathbf{CS}}E_{\mathbf{PR}|\mathbf{NM,CS}}\left(\Delta \mid \mathbf{NM,CS}\right)\right) \\
&\quad + V_{\mathbf{CS}}E_{\mathbf{NM}|\mathbf{CS}}E_{\mathbf{PR}|\mathbf{NM,CS}}\left(\Delta \mid \mathbf{NM,CS}\right) \\
&= E_{\mathbf{CS}}E_{\mathbf{NM}|\mathbf{CS}}V_{\mathbf{PR}|\mathbf{NM,CS}}\left(\Delta \mid \mathbf{NM,CS}\right) + E_{\mathbf{CS}}V_{\mathbf{NM}|\mathbf{CS}}E_{\mathbf{PR}|\mathbf{NM,CS}}\left(\Delta \mid \mathbf{NM,CS}\right) \\
&\quad + V_{\mathbf{CS}}E_{\mathbf{NM}|\mathbf{CS}}E_{\mathbf{PR}|\mathbf{NM,CS}}\left(\Delta \mid \mathbf{NM,CS}\right) \\
&= V(\mathbf{PR}) + V(\mathbf{NM}) + V(\mathbf{CS})
\end{aligned}
\tag{4}
$$





The three terms on the right-hand side of Eq. (4) represent the partial variances contributed from parameter, model and climate scenario, respectively. The equation indicates that the total variance can be decomposed into the variances contributed from the alternative climate scenarios $\mathbf{CS}$, plausible numerical models $\mathbf{NM}$, and uncertain parameters $\mathbf{PR}$. Then, we can define the new sensitivity indices for $\mathbf{PR}$, $\mathbf{NM}$ and $\mathbf{CS}$ following the first-order sensitivity index definition:

$$S_{\mathbf{PR}} = \frac{E_{\mathbf{CS}}E_{\mathbf{NM|CS}}V_{\mathbf{PR|NM,CS}}\left(\Delta \mid \mathbf{NM,CS}\right)}{V\left(\Delta\right)} = \frac{V\left(\mathbf{PR}\right)}{V\left(\Delta\right)}, \tag{5}$$

$$S_{\mathbf{NM}} = \frac{E_{\mathbf{CS}}V_{\mathbf{NM|CS}}E_{\mathbf{PR|NM,CS}}\left(\Delta \mid \mathbf{NM,CS}\right)}{V\left(\Delta\right)} = \frac{V\left(\mathbf{NM}\right)}{V\left(\Delta\right)}, \tag{6}$$

$$S_{\mathbf{CS}} = \frac{V_{\mathbf{CS}}E_{\mathbf{NM,CS}}E_{\mathbf{PR|NM,CS}}\left(\Delta \mid \mathbf{NM,CS}\right)}{V\left(\Delta\right)} = \frac{V\left(\mathbf{CS}\right)}{V\left(\Delta\right)}. \tag{7}$$

**2.2 Sensitivity indices estimation using the Latin hypercube sampling**

In this research, the parameters were sampled within the feasible range via Latin hypercube sampling (LHS) (Kanso et al., 2006; Zhang and Pinder, 2003). LHS is a constrained Monte Carlo sampling, which can accurately reflect the function distribution of the input data. Compared with Monte Carlo sampling, LHS greatly reduces the demand for sample size and

computation time and is widely used in modeling simulation and optimization calculations. For a function: $\mathbf{Y} = f\left(\mathbf{X}\right)$, where $\mathbf{X} = \left\{X_1, X_2, ..., X_3\right\}$, by using LHS, the range of $X_i$, $i = 1, 2, ..., k$ can be divided into $n$ non-overlapping intervals with equal probabilities. The $n$ values obtained from $X_1$ are paired with $n$ values obtained from $X_2$ randomly, these $n$ paired values then combined with those $n$ values from $X_3$. We repeat this process until the new $n \times k$ matrix $\mathbf{X}$ is developed. This

sample matrix $\mathbf{X}$ can be used to calculate the sensitivity index for the model output. More details about LHS are described in previous studies (McKay, 1979; Owen, 1998; Helton, 2003).

Assuming there are $l$ alternative climate scenarios, and under each climate scenario, there are $k$ plausible models and we have $n$ sampled parameter sets for each model and climate scenario. Using the variance definition, the partial variance of $V\left(\mathbf{PR}\right)$ can be expressed as:

$$\begin{aligned} V\left(\mathbf{PR}\right) &= E_{\mathbf{CS}}E_{\mathbf{NM|CS}}V_{\mathbf{PR|NM,CS}}\left(\Delta \mid \mathbf{NM,CS}\right) \\ &= E_{\mathbf{CS}}E_{\mathbf{NM|CS}}\left(E_{\mathbf{PR|NM,CS}}\left(\Delta \mid \mathbf{NM,CS}\right)^2 - \left(E_{\mathbf{PR|NM,CS}}\left(\Delta \mid \mathbf{NM,CS}\right)\right)^2\right). \end{aligned} \tag{8}$$





After applying the formula of expectation and LHS method then the terms of $V(\mathbf{PR})$, $V(\mathbf{NM})$ and $V(\mathbf{CS})$ can be expressed as:

$$
\begin{aligned}
V(\mathbf{PR}) &= E_{\mathbf{CS}} E_{\mathbf{NM|CS}} \left( E_{\mathbf{PR|NM,CS}} \left( \Delta \mid \mathbf{NM}, \mathbf{CS} \right)^2 - \left( E_{\mathbf{PR|NM,CS}} \left( \Delta \mid \mathbf{NM}, \mathbf{CS} \right) \right)^2 \right) \\
&= E_{\mathbf{CS}} E_{\mathbf{NM|CS}} \left( \frac{1}{n} \sum_{j=1}^{n} \Delta^2 \left( PR_j \mid NM_k, CS_l \right) - \left( \frac{1}{n} \sum_{j=1}^{n} \Delta \left( PR_j \mid NM_k, CS_l \right) \right)^2 \right) \\
&= \sum_l \sum_k \left( \frac{1}{n} \sum_{j=1}^{n} \Delta^2 \left( PR_j \mid NM_k, CS_l \right) - \left( \frac{1}{n} \sum_{j=1}^{n} \Delta \left( PR_j \mid NM_k, CS_l \right) \right)^2 \right) P(NM_k \mid CS_l) P(CS_l)
\end{aligned}
\tag{9}
$$

$$
\begin{aligned}
V(\mathbf{NM}) &= E_{\mathbf{CS}} V_{\mathbf{NM|CS}} E_{\mathbf{PR|NM,CS}} \left( \Delta \mid \mathbf{NM}, \mathbf{CS} \right) \\
&= \sum_l P(CS_l) \left( \begin{array}{c} \sum_k \left( \frac{1}{n} \sum_{j=1}^{n} \Delta \left( PR_j \mid NM_k, CS_l \right) \right)^2 P(NM_k \mid CS_l) - \\ \left( \sum_k \left( \frac{1}{n} \sum_{j=1}^{n} \Delta \left( PR_j \mid NM_k, CS_l \right) P(NM_k \mid CS_l) \right) \right)^2 \end{array} \right),
\end{aligned}
\tag{10}
$$

$$
\begin{aligned}
V(\mathbf{CS}) &= V_{\mathbf{CS}} E_{\mathbf{NM,CS}} E_{\mathbf{PR|NM,CS}} \left( \Delta \mid \mathbf{NM}, \mathbf{CS} \right) \\
&= \sum_l P(CS_l) \left( P(NM_k \mid CS_l) \left( \frac{1}{n} \sum_{j=1}^{n} \Delta_k \left( PR_j \mid NM_k, CS_l \right) \right) \right)^2 \\
&\quad - \left( \sum_l \sum_k P(CS_l) P(NM_k \mid CS_l) \left( \frac{1}{n} \sum_{j=1}^{n} \Delta_k \left( PR_j \mid NM_k, CS_l \right) \right) \right)^2
\end{aligned}
\tag{11}
$$

Where $n$ and $j$ represent the parameter LHS sample number and index respectively, $P(NM_k | CS_l)$ represents the weight of model $NM_k$ under climate scenario $CS_l$ with $\sum_k P(NM_k | CS_l) = 1$ and $P(CS_l)$ is the weight of different climate scenario situations satisfying $\sum_l P(CS_l) = 1$. The values of weights for alternative models or climate scenarios could be selected using their prior knowledge or objective criteria, e.g., posterior probabilities of the Bayesian theorem (Neumann, 2012; Schoniger et al., 2014). In this study, we only use the equal prior weights for different models and scenarios considering the calculations of these weights are not the focus of this research.

## 3 Study area and model setup

### 3.1 Site description and data set

The study site locates in the north of the Manaus, Brazil (Fig. 2), with a drainage area of ~9,000 km². Within the central Amazon, the watershed is mostly covered by tropical forest with ~12% cropland and ~5% wetland (based on Community



Land Model (CLM) land surface data; Niu et al., 2017). The average precipitation for this region has large seasonal variability. From December to May is the wet season while from June to November is the dry season. With the relative high elevation (90 – 210 m) of upper landscape and the relative low elevation (45 – 55 m) of swampy valleys, a dense drainage network was formed in the region. The watershed has 4 rivers: Urubu, Preto da Eva, Tarumã-açu, and Tarumã-mirim Rivers.

Various inputs are needed for watershed modeling using PAWS. 90-m resolution NASA Shuttle Rader Topography Mission (SRTM) (U.S. Geological Survey; http://eros.usgs.gov) is applied as DEM input, but for channel network and watershed boundary delineation the Advanced Spaceborne Thermal Emission and Reflection Radiometer (ASTER) provides the 30-m resolution Global Digital Elevation Model version 2 (GDEM V2) instead. Default CLM CRU-NCEP (CRUNCEP) dataset (Piao et al., 2012) serves as the climate forcing of the model, e.g., temperature, humidity, short- and long-wave solar

radiation, wind speed, and land surface air pressure. We replaced the input CRUNCEP precipitation with NASA's Tropical Measuring Mission (TRMM) data (http://trmm.gsfc.nasa.gov/), since the model fails to capture the peak stream discharges using CRUNCEP rainfall data (Niu et al., 2017). CLM land surface data is applied as land use and land cover (LULC) input. More details about the model inputs can be found in Niu et al. (2017).

### 3.2 Model setup

PAWS uses an effective non-iterative scheme to couple hydrologic processes including both land surface and subsurface water. The details of numerical implementation and the governing equations can be found in Shen and Phanikumar (2010) and Niu et al. (2014) (Fig. 3). Briefly, four flow domains are simulated in PAWS, including stream channel, overland flow, vadose zone, and saturated groundwater. Structured grid-based finite-volume method is the main numerical scheme applied to discretize the governing equations in various hydrologic components. PAWS also simulates two land surface subdomains,

i.e., infiltration and evaporation are depicted in the ponding subdomain while the overland flow occurs in the surface flow subdomain. PAWS considers the horizontal interaction of both surface runoff and groundwater flow between the model grids, which represents the actual hydrological processes and is often ignored by other regional and global hydrologic models. The 1D diffusive wave equation is solved to simulate channel flow and the 2D version for overland flow. The leakance concept is the basis applied to simulate the exchange between the channel and groundwater explicitly. PAWS has been coupled with

CLM (Shen et al., 2013), which calculates the surface energy balance and soil and plant carbon and nitrogen cycles. Canopy interception and *ET* demand (both transpiration and soil evaporation) are also computed in CLM at each time step.

A 1 km × 1 km grid was used for horizontal discretization, resulting in 118 × 122 grid cells for the study site. To discretize the vadose zone 20 vertical layers were used, and for the fully saturated groundwater there were two layers-the unconfined aquifer on the top layer while the confined aquifer on the bottom. The depth from water table to bedrock was fixed as 100

meters. The initial groundwater head was obtained from the global groundwater table depth data (Fan et al., 2013) for the hydrologic component spin-up. From the initial carbon and nitrogen states, an uncoupled version of CLM (i.e., not coupled with PAWS) was run for 525 simulation years; the resulting states were then used to initialize the PAWS+CLM coupled



simulation. The spun-up states were verified through analysis of leaf area index (LAI). The model calibration and testing use multiple datasets, including the stream discharge observations at two stream gauges (15042000 in Preto da Eva River and

16010000 in Urubu River; Fig. 1 in Niu et al. (2017)) from the Agência Nacional de Águas (ANA) Hidroweb Sistema de Informações Hidrológicas (http://hidroweb.ana.gov.br/), MODIS monthly *ET* (MODIS 16A2) data (Mu et al., 2007, 2011), and GRACE land grid equivalent water thickness data (http://geoid.colorado.edu/grace/dataportal.html). The details of these data can be referred to Niu et al. (2017).

The Tropical Measuring Mission (TRMM) 3B42 daily V7 data from NASA (available at http://trmm.gsfc.nasa.gov/), which

has 0.258 spatial resolution, was used as weather data in this model. Two simulations extend periods from 1 September 1999, 1:00 AM to 1 June 2000, 1:00 AM and from 1 March 2000, 1:00 AM to 27 November 2000, 1:00 AM each with a duration of 6,528 hours (272 days, ~9 months). We chose the climate datasets of the years of 1999 and 2000 because it avoids severe drought or flood situations which will help us better understand the hydrology of the normal dry and wet seasons in the central Amazon. More detailed information about data input and simulation details are described in Niu et al. (2017).

**3.3 Uncertainty sources**

There are three main uncertainty sources for the hydrologic processes modeling by using PAWS+CLM in the central Amazon: (1) climate scenarios, including the precipitation, radiation intensity and temperature at different seasons; (2) the conceptualization of numerical model, i.e., the optional overland flow module; and (3) parameters. These uncertainty sources follow the hierarchical structure shown in Fig. 1. The model simulation time was 6,528 hours under each climate scenario.

The first 2,208 hours (92 days, ~3 months) were regarded as initialization and the remaining 4,320 hours (180 days, ~6 months) were used for analysis. One simulation period of 4,320 hours from 2 December 1999, 1:00 AM to 29 May 2000, 00:00 PM was established as climate scenario $CS_1$, while the other simulation period of 4,320 hours from 1 June 2000, 1:00 AM, to 27 November 2000, 00:00 PM was established as climate scenario $CS_2$. $CS_1$ represents the wet season with average precipitation of 10.8 mm day$^{-1}$, average radiation intensity of 544.1 MJ day$^{-1}$m$^{-2}$, and average temperature of 27.1℃, while

$CS_2$ represents the dry season, with average precipitation of 5.0 mm day$^{-1}$, average radiation intensity of 589.8 MJ day$^{-1}$m$^{-2}$, and average temperature of 27.8℃. In general, the scenario $CS_1$ has more frequent heavy rains and weaker radiation intensity comparing with scenario $CS_2$ (Fig. 4).

Two alternative conceptual models, consisting of different PAWS modules, were considered in this study. One model contains the overland flow module while the other model excludes the overland flow module. Therefore, there are two

models ( $NM_1$ and $NM_2$ ) under each climate scenario. In this study, we used equal weights for the different models and climate scenarios, leading to $P(CS_l) = P(NM_k | CS_l) = 0.5$ .



We adjusted the values of five model parameters to explore total parameter sensitivity: Van Genuchten soil parameters $\alpha$ and $N$, soil saturated conductivity $K_s$, unconfined aquifer conductivity $K_1$, and confined aquifer conductivity $K_2$ (Table 1). In order to make sensitivity analysis uncomplicated and computationally tractable, each grid was taken the same value for a

specific parameter. The soil saturated conductivity $K_s$ ($m^2day^{-1}$), unconfined aquifer conductivity $K_1$ ($m^2day^{-1}$), and confined aquifer conductivity $K_2$ ($m^2day^{-1}$) were assumed to follow lognormal distributions (log-N (1.6094, 0.4214$^2$), log-N (3.4012, 0.4214$^2$), and log-N (1.6094, 0.4214$^2$), respectively). The lower bounds of these three lognormal distributions were all zero and the higher bounds were 10, 60, and 10, respectively. The remaining two parameters (VG parameters $\alpha$ and $N$) were assumed to follow a uniform distribution: U (0.1, 4) and U (1.03, 5), respectively. This set of parameters was used in each

model under each climate scenario.

## 4 Results and discussion

### 4.1 Model predictions

The total number of PAWS+CLM simulations considering all possible combinations of three uncertain factors was $2\times2\times100=400$, which represents two climate scenarios, two model conceptualizations (with and without overland flow

module), and 100 sampled parameter sets. The results of these 400 realizations for three outputs of interest: $ET$, $E_G$ and $Q_G$ are summarized in Fig. 5, and demonstrate that the predictions of $ET$, $E_G$, and $Q_G$ all exhibited substantial uncertainty that varied across different time steps. The results also demonstrate that the overland flow significantly influences $Q_G$ predictions because the simulation results without the overland flow module (red and yellow dots in Fig.4) are higher than those with overland flow module (blue and green dots).

### 4.2 Sensitivity indices for evapotranspiration

We first calculated the sensitivity indices for the spatial averaged $ET$ of the whole watershed at all time steps using Eqs. (5) - (7). In this study, one time-step represents one hour. Six time points (the simulation times = 1777 h, 1785 h, 2521 h, 2529 h, 3265 h, and 3273 h) were chosen as examples to show sensitivity indices (Fig. 6). Simulation times of 1777 h, 2521 h, and 3265 h, which belonged to different days, but all correspond to 1:00 AM local time. At these time points, the model

uncertainty ($S_{NM}$) was the most important contributor to the total $ET$ prediction uncertainty, accounting for 46-66% of total uncertainty, and climate scenarios ($S_{CS}$) contributed the least uncertainty. However, at different time points of (1785 h, 2529 h, and 3273 h, corresponding to 9:00 AM local time), the climate scenarios were the dominant uncertainty contributor with $S_{CS}$ ranging from 74-90%. These results indicate that the sensitivity to various factors is strongly temporally dependent (Fig. 7 shows the $ET$ prediction sensitivity indices at 1:00 AM, 5:00 AM, 9:00 AM, 1:00 PM, 5:00 PM, and 9:00 PM local time).

All the sensitivity indices fluctuate strongly with time and there is no clear periodic information in these values (Fig. 7). Furthermore, the results exhibit a regular pattern that climate scenarios generally were the most important uncertainty source



total $Q_G$ prediction uncertainty, while the uncertainty contribution of climate scenarios is negligible. This pattern can be explained by the fact that the climate scenarios related variables having no direct immediate influence on groundwater flow. These results are consistent with the previous study of Niu et al. (2017) that groundwater has buffering effects on climate

impacts from storms and droughts in the central Amazon. Comparing with the results of $E_G$, the maximum values of $S_{NM}$ occur at a later time (5:00 AM) which implies a longer time lag after precipitation because of the slow infiltration process from soil to aquifer.

Because groundwater exchange with stream flow occurs only at grid cells along the streams, the sensitivity indices only having valid values in those stream grid cells (Fig. 12). Our results indicate that the model parameters are the most important

contributor to total uncertainty of time averaged $Q_G$ predictions. At some grid cells, the model uncertainty dominates, implying that overland flow is important at these locations.

**5 Conclusions**

In this study, we implemented an advanced hierarchical sensitivity analysis method into a complex large-scale process-based hydrological model (PAWS) to identify important uncertain inputs for $ET$, $E_G$, and $Q_G$ predictions in an Amazonia watershed.

This sensitivity analysis method is capable of providing accurate measurements for importance of uncertain inputs through variance decomposition and it can also categorize and combine different uncertain inputs considering their dependence relationship to decrease the high dimensionality induced by the complex and large-scale problem. Three groups of uncertainty sources or uncertain inputs were considered in this research, including alternative climate scenarios, two plausible model structures, and uncertain model parameters (i.e., soil parameter, soil saturated conductivity, unconfined

aquifer conductivity, and confined aquifer conductivity). A hierarchical framework of uncertainty and a new set of sensitivity indices were defined for these uncertainty sources. The Latin hypercube sampling method was implemented to calculate the sensitivity indices to save the computational cost.

The sensitivity analysis results in this research demonstrate strong temporal and spatial variations, and there is no single dominant uncertainty class for different model predictions. The climate scenarios are the most important uncertainty

contributor for spatially-averaged $ET$ predictions during daytime, while parameter uncertainty is the most important factor during night time. In contrast, parameter uncertainty is the most important contributor to total uncertainty of spatially-averaged $E_G$ and $Q_G$ predictions at all times. For the time averaged $ET$ and $E_G$ predictions, parameters are also the most important uncertainty source at along streams, while the model uncertainty dominates at other areas in the watershed. For time-averaged $Q_G$ predictions, parameter uncertainty dominates at most stream grid cells. In general, model parameters are

the most important uncertainty contributor for all model predictions, especially those within streamside model grid cells. In addition, the overland flow module induced model uncertainty plays a non-ignorable influence in model predictive uncertainty. The distinct sensitivity analysis result patterns of different uncertain inputs can be explained by different physical processes involved in the conceptual model.





We demonstrated a pilot example for comprehensive global sensitivity analysis of large-scale hydrological models. The
sensitivity analysis results can provide key information of model uncertainty sources for modelers and inform needed
measurements and process representation improvements. We note that the proposed method is mathematically rigorous and
general and can be applied to extensive large scale hydrological or environmental models with more or different sources of
uncertainty.

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





**Table 1: Five chosen parameters to be included in parameter uncertainty**

| Parameters | Units | Description | Allowable Range |
|---|---|---|---|
| $K_s$ | m day$^{-1}$ | soil saturated hydraulic conductivity | 0.0-10.0 |
| $K_1$ | m day$^{-1}$ | unconfined aquifer hydraulic conductivity | 0.0-60.0 |
| $K_2$ | m day$^{-1}$ | confined aquifer hydraulic conductivity | 0.0-10.0 |
| $\alpha$ | | Van Genuchten parameter | 0.1-4.0 |
| $N$ | | Van Genuchten parameter | 1.03-5.0 |


**Figure 1: The hierarchical sensitivity analysis framework developed for PAWS applied in the central Amazon basins. The three uncertainty sources are placed into the appropriate hierarchy level according to their dependence relationships.**





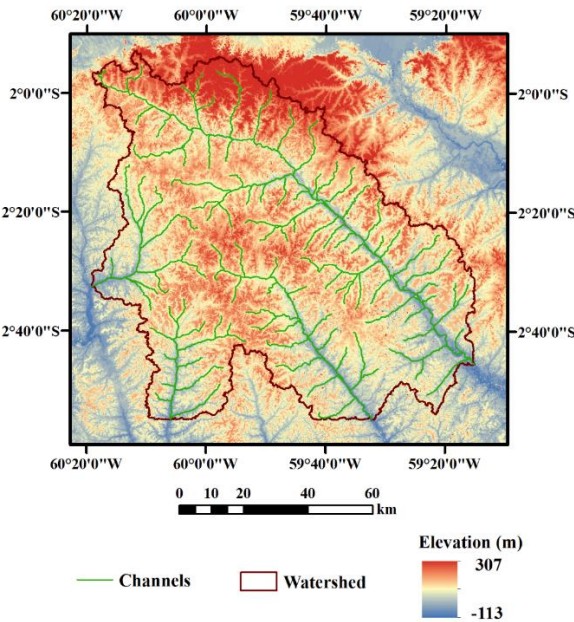


Figure 2: The 2-D map of watershed used in this study, showing the elevation, channels and watershed boundary. The study area extends from 1°57′36″S to 2°56′0″W and 59°14′48″W to 60°20′0″W.

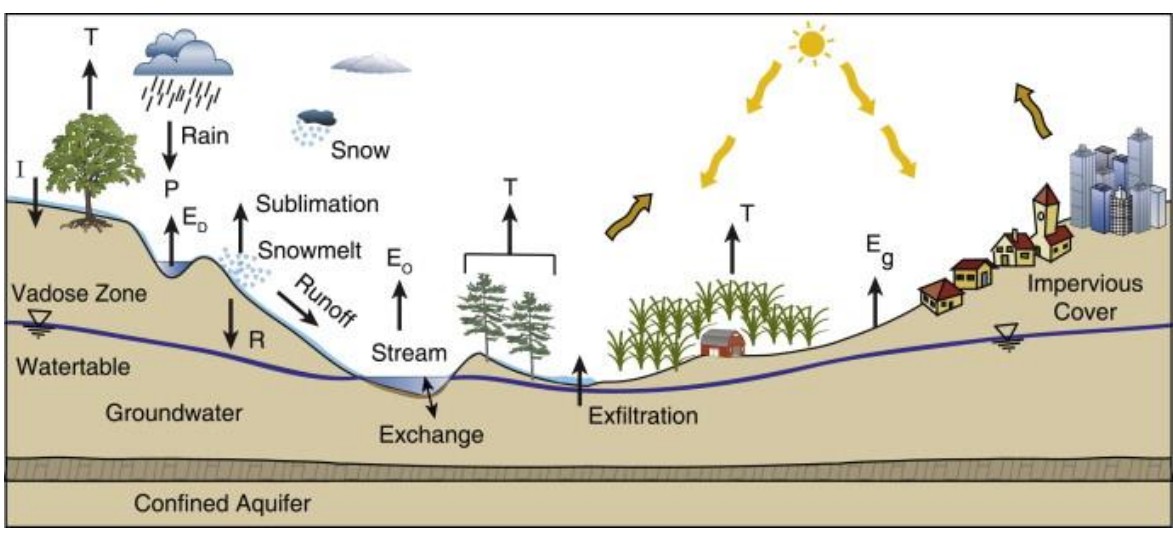

Figure 3: A schematic of the processes represented in PAWS (Shen and Phanikumar, 2010). T: transpiration, P: precipitation, $E_g$: evaporation from bare soil, $E_O$: evaporation from overland flow/stream, $E_D$: evaporation from depression storage, I: infiltration; R: recharge.



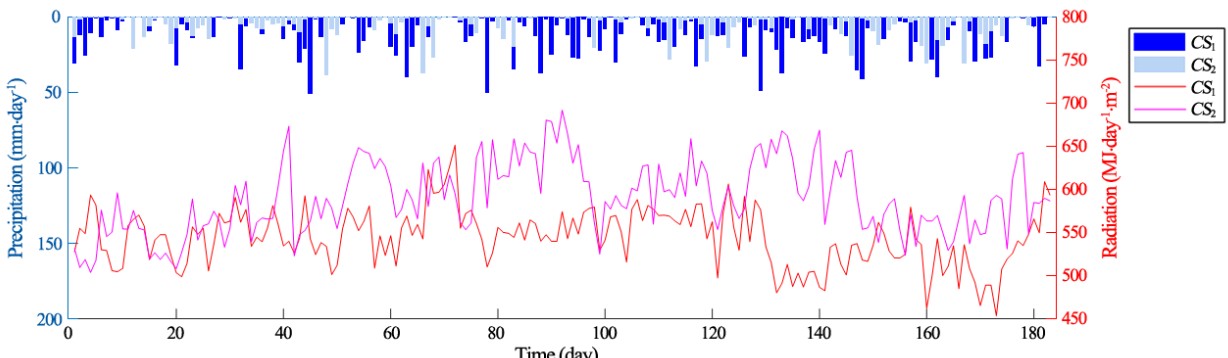

**Figure 4: Precipitation (from TRMM 3B42 V7 precipitation data) and solar radiation (from CRUNCEP climate forcing data) forcing of two different climate scenarios $CS_1$ and $CS_2$. Average precipitation of $CS_1$ is 10.8 mm day$^{-1}$ and average radiation intensity is 544.1 MJ day$^{-1}$m$^{-2}$. $CS_2$ represents the dry season, with average precipitation of 5.0 mm day$^{-1}$, average radiation intensity of 589.8 MJ day$^{-1}$m$^{-2}$.**




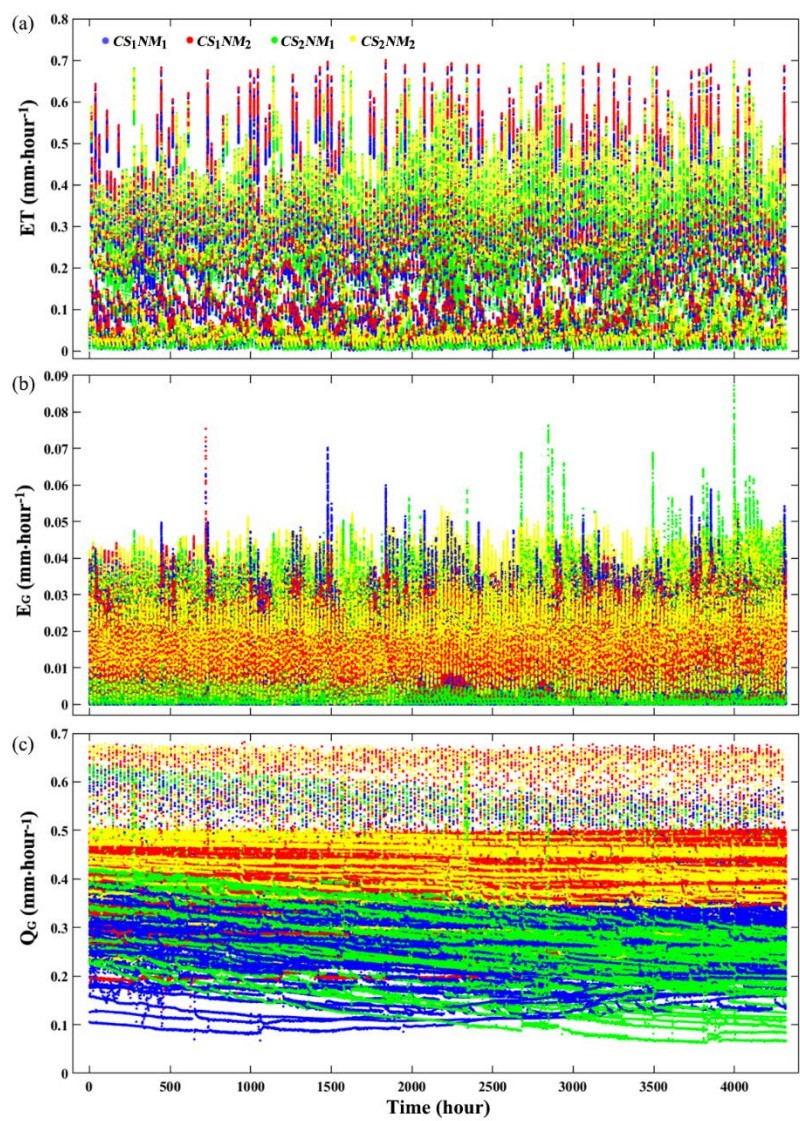

**Figure 5:** *ET* (a), *E_G* (b), and *Q_G* (c) for 4320 hours simulation period (180 days, ~6 months) for all simulations. The blue, red, green, and yellow dots represent the results under wet season and with overland flow module, wet season without overland flow module, dry season with overland flow, and dry season without overland flow, respectively. Since there is 100-size sample of parameters under a fixed numerical model under a fixed climate scenario, each hour possesses 400 dots (100 dots of each color) in total. All results of *ET*, *E_G*, and *Q_G* exhibited substantial uncertainty caused by climate scenario, overland flow and parameters.





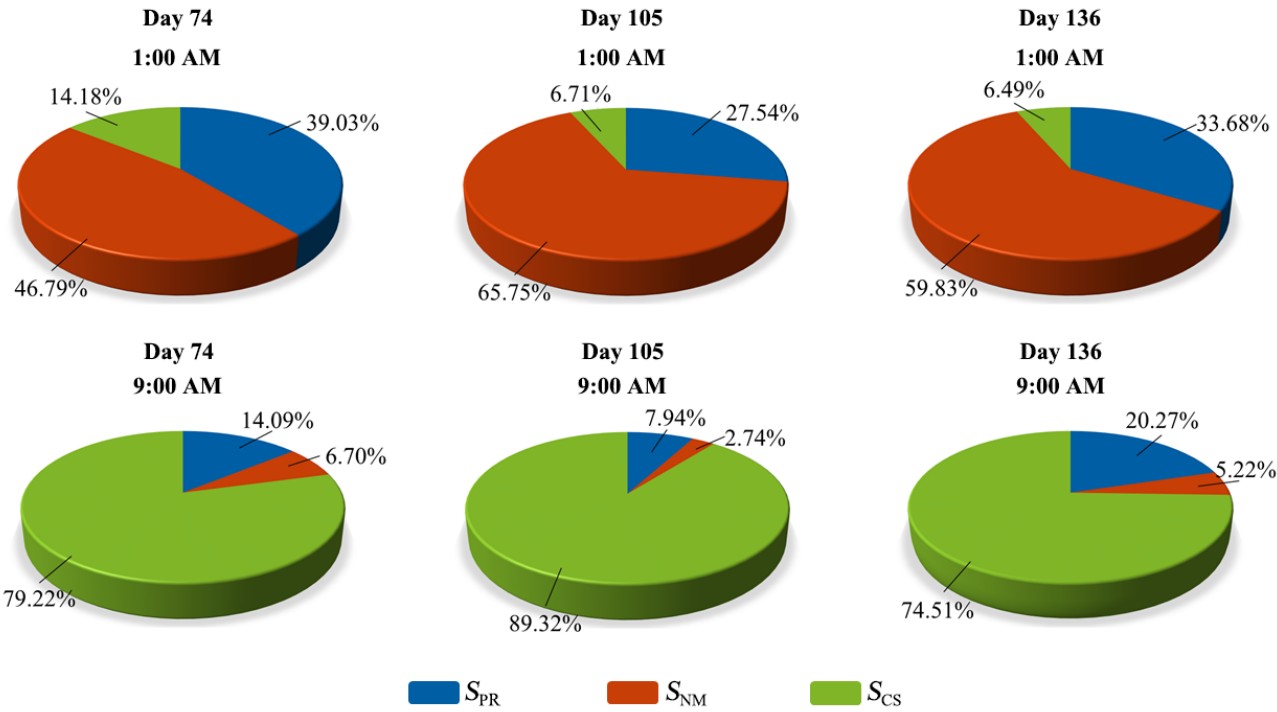

**Figure 6: Estimated sensitivities for the spatial averaged *ET* at 6 time points (the simulation times = 1777 h (Day 74, 1:00 AM), 1785 h (Day 74, 9:00 AM), 2521 h (Day 105, 1:00 AM), 2529 h (Day 105, 9:00 AM), 3265 h (Day 136, 1:00 AM), and 3273 h (Day 136, 9:00 AM)).**




Figure 7: $S_{PR}$, $S_{NM}$ and $S_{CS}$ of the spatial averaged *ET* predictions at (a) 1:00 AM (b) 5:00 AM (c) 9:00 AM (d) 1:00 PM (e) 5:00 PM and (f) 9:00 PM extracted from all simulations. All the sensitivity indices fluctuate strongly with time and there is no clear periodic information in these values.

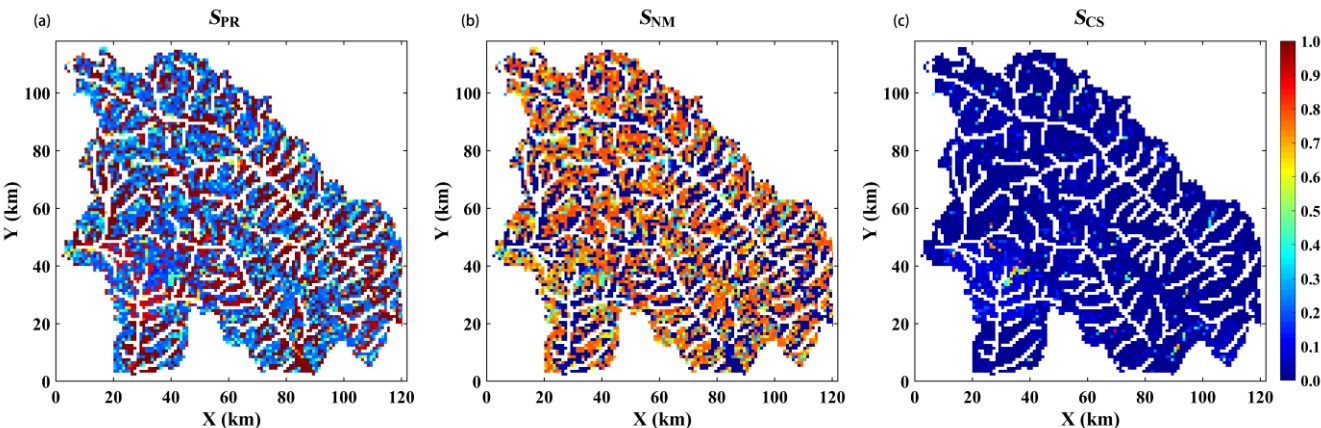

**Figure 8:** The maps of (a) $S_{PR}$, (b) $S_{NM}$ and (c) $S_{CS}$ for mean value of *ET* predictions over the whole simulation period (the initialization period is excluded) exhibit a spatially determined pattern: *ET* is most sensitive to parameters within the stream grid cells while within the grid cells away from stream, *ET* is most sensitive to overland flow.


**Figure 9:** $S_{PR}$, $S_{NM}$ and $S_{CS}$ of the spatial averaged $E_G$ predictions at (a) 1:00 AM (b) 5:00 AM (c) 9:00 AM (d) 1:00 PM (e) 5:00 PM and (f) 9:00 PM extracted from all simulations. At 9:00 AM, 1:00 PM, 5:00 PM and 9:00 PM, the parameters are the most important uncertainty contributor. At night (1:00 AM, 5:00 AM and 9:00 PM) $E_G$ is very sensitive to model uncertainty. The climate scenarios are unimportant for $E_G$ predictions.



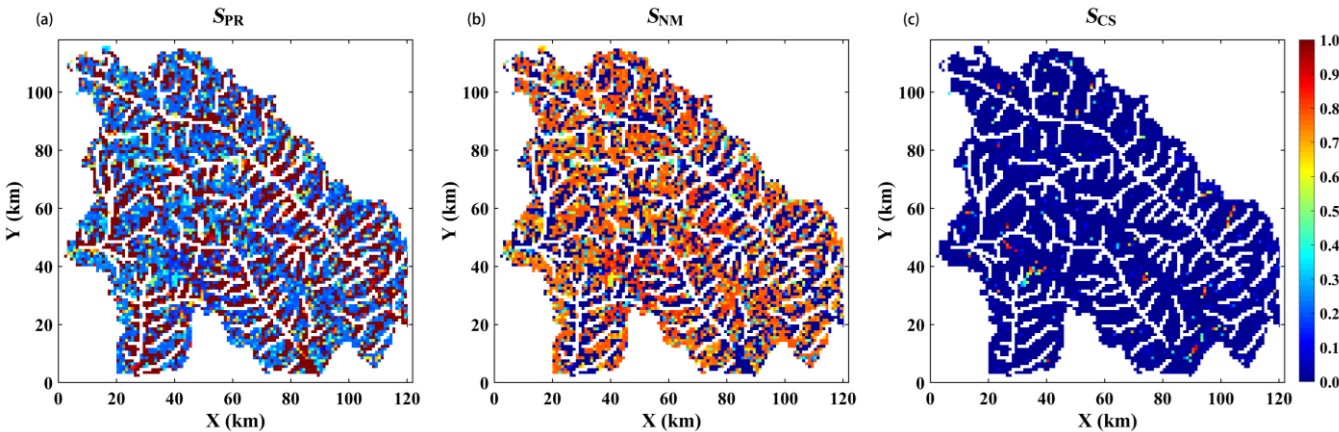


**Figure 10: The maps of (a) $S_{PR}$, (b) $S_{NM}$ and (c) $S_{CS}$ for mean value of $E_G$ predictions over the whole simulation period (the initialization period is excluded). They exhibit a similar pattern to $ET$ (Fig. 8).**

**Figure 11:** $S_{PR}$, $S_{NM}$ and $S_{CS}$ of the spatial averaged $Q_G$ predictions at (a) 1:00 AM (b) 5:00 AM (c) 9:00 AM (d) 1:00 PM (e)
**5:00 PM and (f) 9:00 PM extracted from all simulations. It indicates that model parameters are the dominant contributor to total**
$Q_G$ **prediction uncertainty while the uncertainty contribution of climate scenarios is negligible.**





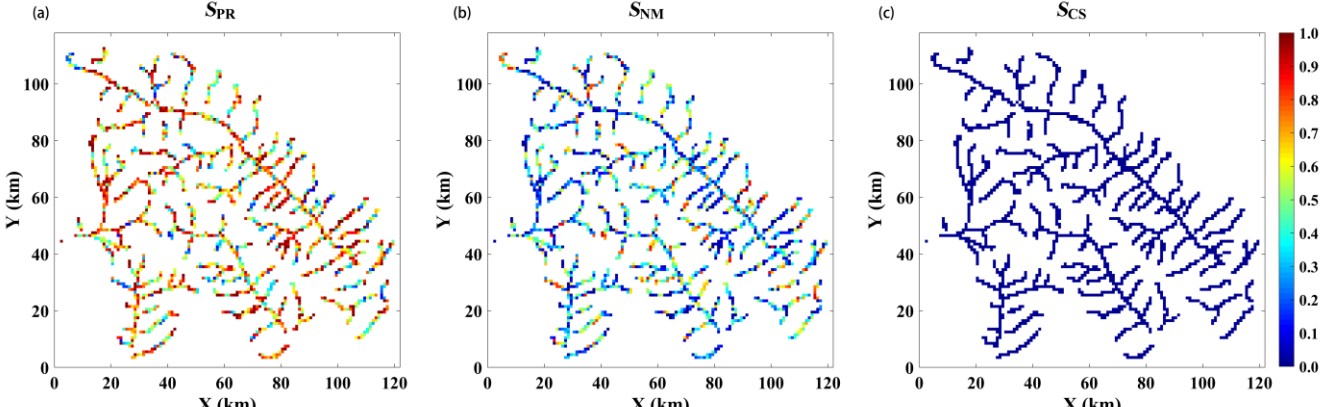

**Figure 12: The patterns of (a) $S_{PR}$, (b) $S_{NM}$ and (c) $S_{CS}$ for mean value of $Q_G$ predictions over the whole simulation period (the initialization period is excluded). They suggest the model parameters are the most important contributor.**
