# Peer review of "Hierarchical Sensitivity Analysis for Large Scale Process-based Hydrological Modeling with Application in an Amazonian Watershed"

_Hydrology and Earth System Sciences, 2019_

## Referee Comment (RC1) · Anonymous Referee #1 · 10 Sep 2019

The topic of the paper is of potential interest for HESS readers. However, I found the paper confused regarding both the exposition of the study and the methodological framework. I suggest a complete revision of the presentation. Some specific comments are as follows: 1) Uncertainties includes climate scenarios (external source), the model itself and some of its parameters. However, considering 2 alternatives for the model and climate scenario does not mean that these are uncertain. Are these alternatives associated with different probabilities? It seems that when the model and climate scenario are fixed uncertainty reduces to parametric uncertainty. I think the authors should explain better how the proposed approach is able to really cover all the sources of uncertainty. 2) To me, uncertainty in climate scenarios should be linked to

uncertainty in climate variables, that should be treated as random variables or random fields. Here, it seems that climate time series are only used to establish two deterministic scenarios. I don't think this is a remarkable innovation for the research in this field. 3) I'm not sure that analysis of variance is the right tool to tackle modeling uncertainty. One could combine GSA with e.g. model discrimination criteria to effectively understand the diverse performance of different models. 4) The discussion of the results is very weak and does not help understanding the physical findings that the study brings.

---

## Referee Comment (RC2) · Anonymous Referee #2 · 25 Sep 2019

General notes

The topic is interesting and relevant for the scope of HESS. However the paper does not clarify if there are new significant results achieved and the innovations are not clear. In particular hierarchical global sensitivity analysis was already implemented by Dai & Ye (2015), several applications are reported in literature (see for example Dai et al., 2017; Dai et al., 2019) and the new contribution with this paper to the technique remains unclear. The suggestion is a deeper analysis of the work and therefore it should be completely reviewed.

Specific comments

1. Page 3 Line 68 - Hierarchical global sensitivity is not implemented for the first time. New aspects are related to parameter sampling technique and the general framework is applied to PAWS+CLM hydrological model for the first time. This is not explained inside the paper.

2. Page 4 Line 98 - Parameter $\alpha$ should have the dimension of the inverse of a length $(L^{-1})$ and not dimensionless. Physical meaning of the Van Genuchten parameters used for the sensitivity analysis should be reported.

3. Page 7 Line 175 - Prior weights for models and scenarios may affect output results. The choice of equal weights should be motivated. An interesting point might be studying the variability of results with respect to different weights. This could be a useful tool to understand different sources of uncertainty.

4. Page 10 Line 245 – Conductivities K have wrong measurement units.

5. Page 10 Line 255 – It is not clear which outputs are reported in Figure 5, if they are spatial averaged or not. Comments to Figure 5c needs a more detailed explanation (and figure reference is not 4 but 5).

6. Page 11 Line 290 - An appendix with main model equations should be included for reader understanding.

7. Figure captions are too short and only acronyms of variables are reported. They should be more exhaustive.

8. Formulas and indices need references.

9. Physical interpretation of results is very poor and absent in general. Substantial conclusions of the work are not highlighted. It is not clear if the hierarchical sensitivity analysis is a good tool to capture output sensitivity related to several uncertainties or not.

References

Dai, H., Ye, M. (2015). Variance-based global sensitivity analysis for multiple scenarios and models with implementation using sparse grid collocation. Journal of Hydrology, 528, 286-300

Dai, H., Chen, X., Ye, M., Song, X., Zachara, J. M. (2017). A geostatistics‐informed hierarchical sensitivity analysis method for complex groundwater flow and transport modeling. Water Resources Research, 53(5), 4327-4343

Dai, H., Ye, M., Hu, B. X., Niedoroda, A. W., Zhang, X., Chen, X., ... Niu, J. (2019). Hierarchical sensitivity analysis for simulating barrier island geomorphologic responses to future storms and sea-level rise. Theoretical and Applied Climatology, 136(3-4), 1495-1511

---

## Author Comment (AC1) · 23 Jan 2020

We greatly appreciate the reviewer for his/her valuable comments and feedback on our study. We have substantially revised the manuscript to address the criticisms, and the main revisions are summarized below:

1.Reviewer 1 made a major comment about the weakness of using only two synthetic climate scenarios and indicated that the scenarios are too simple without considering climate variables. To address this comment, we have totally changed the alternative scenarios used in this study. The number of different scenarios has increased from 2 to 6, and they are all now generated from real climate data. The generation procedure

for the new scenarios can be described as follows: the annual weather data from 1998 to 2013 were first collected and divided into multiple dry and wet seasons. Then, these seasons were sorted according to their total precipitation values, and they were divided into six different groups representing six climate scenarios from wet to dry (three for the dry season and three for the wet season). The mean and standard deviation values of the different climate variables (e.g., precipitation, maximum temperature) for each group were further calculated using their daily climate data. Finally, we generated random daily climate data for each climate scenario based on these mean and standard deviation values using the normal distribution.

2.Reviewer 1 raised a question on the generation of model uncertainty. In the revised manuscript, to address the comment, we have changed the uncertainty of the model from with/without the overland flow module to that associated with three plausible aquifer models. This revision adds physical meaning to model uncertainty because when establishing the hydrological model for the research area, the thicknesses of the different aquifers were uncertain and can be described by different conceptual models. According to Pellertier et al. (2016), the thickness of the soil and unconsolidated rocky material exceeded the maximum value in their model (50 m) in the central Amazon region; therefore, we built three aquifer models considering different thicknesses for the unconfined and confined aquifers: (1) 100 m and 200 m, (2) 50 m and 250 m, and (3) 250 m and 50 m, respectively.

3.Both reviewers commented about deficiencies in terms of the results and deep discussion, especially the absence of physical interpretation. To solve this problem, we have totally revised the uncertainty sources, including all uncertain inputs of scenarios, models, and parameters, used in this study. The generation of new climate scenarios and plausible models has been described above, and the parametric uncertainty has also been totally revised. One new uncertain parameter, which is the length of the flow path for runoff contribution to the overland flow domain, has been added. The six parameters were further divided into three groups: vadose zone parameters, groundwater parameters, and the overland flow parameter. A new set of subdivided parametric sensitivity indices was further calculated for each parameter group. To implement the sensitivity analysis, we have performed a more physical and practical interpretation of the model parameters and structures. To estimate the subdivided parametric sensitivity indices, we implemented the Latin hypercube sampling method and binning method with the hierarchical sensitivity analysis method for the first time. The new sensitivity analysis results can provide more detailed information on the importance of different uncertainty sources for modelers. The size of the parameter samples was also increased from 100 to 600. Therefore, the total number of simulations increased from "$2 \times 2 \times 100=400$" to "$6 \times 3 \times 600=10,800$".

4.Both reviewers commented on the equal weights (probabilities) used for the alternative scenarios and models. To address the comment, we have added a section to the revised manuscript to discuss the new sensitivity analysis results using different weight values for the scenarios and models.

5.Reviewer 2 commented on the description of the governing equations for the PAWS model. To address this comment, we have added an appendix to describe the governing equations and parameters of PAWS in detail.

General Evaluation

The topic of the paper is of potential interest for HESS readers. However, I found the paper confused regarding both the exposition of the study and the methodological framework. I suggest a complete revision of the presentation.

Response

We thank the reviewer for the positive evaluation of this manuscript and constructive comments. To improve the presentation of this study and implement the methodology, we have completely revised the framework of the description of the methodology, study area, and model setup. First, we described the general condition of the study area and

the PAWS model with implementation information for PAWS+CLM used in this study. Next, we introduced the hierarchical sensitivity analysis method and its implementation. Then, we described the Latin hypercube sampling method and binning method used to calculate the sensitivity indices. Finally, we introduced detailed information on the generation of the different uncertain inputs considered in this study.

Comment 1

Uncertainties includes climate scenarios (external source), the model itself and some of its parameters. However, considering 2 alternatives for the model and climate scenario does not mean that these are uncertain. Are these alternatives associated with different probabilities? It seems that when the model and climate scenario are fixed uncertainty reduces to parametric uncertainty. I think the authors should explain better how the proposed approach is able to really cover all the sources of uncertainty.

Response

We understand the reviewer's confusion, and we have totally revised the generation of the scenarios and model uncertainties. We have increased the number of alternative scenarios from two to six in the way described in the above summary of the main revisions. The plausible models have also been changed from models with/without an overflow module to three different aquifer models with physical meaning. The scenario and model uncertainties are induced by these coexisting scenarios and models. Figure 1 provides a synthetic example of probability density functions (PDFs) for the model predictions under multiple models and scenarios. The between-scenario variance (the term in Eq. (2)) represents the scenario uncertainty for this synthetic example. As the figure shows, the scenario and model uncertainties represent the distinct predictions provided by different coexisting models and scenarios, and they exist with or without parametric uncertainty. Our methodology used the variance decomposition technique to measure these uncertainties and separated them from the scenarios, then models, then parameters following their dependence relationships (Equations (2) to (7)). All the

uncertainty sources are indeed covered and accurately measured in our study through this uncertainty framework and variance decomposition system.

The alternative scenarios and models can indeed have different probabilities or weights, as the reviewer mentioned. We have added a section (section 3.5) to investigate the influences of different prior probabilities on the climate scenarios and numerical models. We changed the prior probabilities for the first numerical model (i.e., the original aquifer model in Niu et al., 2017) and the prior probabilities for the wettest and driest climate scenarios. The results are presented in Section 3.5. We also changed Figure 2 (the hierarchical sensitivity analysis framework developed for PAWS applied in the central Amazon basins) to clarify the approach we proposed in this study.

Comment 2

To me, uncertainty in climate scenarios should be linked to uncertainty in climate variables, that should be treated as random variables or random fields. Here, it seems that climate time series are only used to establish two deterministic scenarios. I don't think this is a remarkable innovation for the research in this field.

Response

We agree with the reviewer. We have totally revised our strategy for defining a climate scenario. As we indicate in the summary of the main revisions, we analysed 16 years of annual weather data in this study area and generated six different climate scenarios. The random climate variables (e.g., precipitation, temperature, radiation intensity, humidity and wind speed) for each scenario were all randomly generated based on the statistical data for the corresponding season. We summarized the statistical information for six climate variables, which is displayed in Table 1.

Comment 3

I'm not sure that analysis of variance is the right tool to tackle modeling uncertainty.

One could combine GSA with e.g. model discrimination criteria to effectively understand the diverse performance of different models.

Response

We understand the reviewer's confusion. This variance-based hierarchical sensitivity analysis method is not the only way to tackle the modelling of uncertainty. However, the variance-based sensitivity analysis method is one of the most widely used and robust GSA methods for accurately quantifying different uncertainties and has no requirement for model characteristics (Yang, 2011). This hierarchical variance-based sensitivity analysis method we used can combine all the uncertainty sources together and provide a set of accurate measurements of their importance. This method thus provides a valid and efficient way to compare model uncertainty with other uncertainties (i.e., scenario and parametric). To the best of our knowledge, this methodology is the only tool available to achieve this goal. The model discrimination criteria mentioned by the reviewer indeed represent an important tool for evaluating and ranking different plausible models. However, they cannot provide measurements of the total model uncertainty or compare the model uncertainty with other uncertainty sources. If the criteria were implemented in this study, it would indeed help us to provide more accurate weights or probabilities for the different models. However, how to best estimate the model criteria and assign the weights is still an open research question and not the focus of this study. Therefore, we added section 3.5 to discuss the effects of using different model weights on the sensitivity analysis but without using the model discrimination criteria. We will further investigate this topic in future research.

Comment 4

The discussion of the results is very weak and does not help understanding the physical findings that the study brings.

Response

We agree with the reviewer. We have totally revised the uncertainty framework and uncertain inputs in our study to make them have more physical meaning. Accordingly, we have revised our results and discussion based on the new findings.

Figure 1. A Demonstration of within- and between-scenario variances using the probability density functions (PDFs) of model predictions.

Figure 2. The framework of the hierarchical sensitivity analysis developed for PAWS applied to the central Amazon basins. The three uncertainty source types are placed into the appropriate hierarchical level according to their dependence relationships. The left part of this figure shows the sources of these uncertainties, and the right side shows the abbreviations and the structural relationships among the various uncertainties. The number of climate scenarios in this study is six; the number of plausible numerical models under each climate scenario is three; and the number of parameter sets under each numerical model is 600. It should be noted that the parameter uncertainty sources are further divided into three parts: vadose zone parameters, groundwater parameters and the overland flow parameter.

References

Niu, J., Shen, C., Chambers, J. Q., Melack, J. M., & Riley, W. J. (2017). Interannual variation in hydrologic budgets in an amazonian watershed with a coupled subsurface - land surface process model. Journal of Hydrometeorology, JHM-D-17-0108.1.

Pelletier, J. D., Broxton, P. D., Hazenberg, P., Zeng, X., Troch, P. A., & Niu, G. Y., et al. (2016). A gridded global data set of soil, intact regolith, and sedimentary deposit thicknesses for regional and global land surface modeling. Journal of Advances in Modeling Earth Systems, 8(1), 41-65.

Yang, J. (2011). Convergence and uncertainty analyses in Monte-Carlo based sensitivity analysis. Environmental Modelling & Software, 26(4), 444-457.

Please also note the supplement to this comment:
https://www.hydrol-earth-syst-sci-discuss.net/hess-2019-246/hess-2019-246-AC1-supplement.pdf

————————————————————

[Figure]

[Figure]

**Fig. 1.**

[Figure]

**Fig. 2.**

---

## Author Comment (AC2) · 23 Jan 2020

We greatly appreciate the reviewer for his/her valuable comments and feedback on our study. We have substantially revised the manuscript to address the criticisms, and the main revisions are summarized below:

1.Reviewer 1 made a major comment about the weakness of using only two synthetic climate scenarios and indicated that the scenarios are too simple without considering climate variables. To address this comment, we have totally changed the alternative scenarios used in this study. The number of different scenarios has increased from 2 to 6, and they are all now generated from real climate data. The generation procedure

for the new scenarios can be described as follows: the annual weather data from 1998 to 2013 were first collected and divided into multiple dry and wet seasons. Then, these seasons were sorted according to their total precipitation values, and they were divided into six different groups representing six climate scenarios from wet to dry (three for the dry season and three for the wet season). The mean and standard deviation values of the different climate variables (e.g., precipitation, maximum temperature) for each group were further calculated using their daily climate data. Finally, we generated random daily climate data for each climate scenario based on these mean and standard deviation values using the normal distribution.

2.Reviewer 1 raised a question on the generation of model uncertainty. In the revised manuscript, to address the comment, we have changed the uncertainty of the model from with/without the overland flow module to that associated with three plausible aquifer models. This revision adds physical meaning to model uncertainty because when establishing the hydrological model for the research area, the thicknesses of the different aquifers were uncertain and can be described by different conceptual models. According to Pellertier et al. (2016), the thickness of the soil and unconsolidated rocky material exceeded the maximum value in their model (50 m) in the central Amazon region; therefore, we built three aquifer models considering different thicknesses for the unconfined and confined aquifers: (1) 100 m and 200 m, (2) 50 m and 250 m, and (3) 250 m and 50 m, respectively.

3.Both reviewers commented about deficiencies in terms of the results and deep discussion, especially the absence of physical interpretation. To solve this problem, we have totally revised the uncertainty sources, including all uncertain inputs of scenarios, models, and parameters, used in this study. The generation of new climate scenarios and plausible models has been described above, and the parametric uncertainty has also been totally revised. One new uncertain parameter, which is the length of the flow path for runoff contribution to the overland flow domain, has been added. The six parameters were further divided into three groups: vadose zone parameters, groundwater parameters, and the overland flow parameter. A new set of subdivided parametric sensitivity indices was further calculated for each parameter group. To implement the sensitivity analysis, we have performed a more physical and practical interpretation of the model parameters and structures. To estimate the subdivided parametric sensitivity indices, we implemented the Latin hypercube sampling method and binning method with the hierarchical sensitivity analysis method for the first time. The new sensitivity analysis results can provide more detailed information on the importance of different uncertainty sources for modelers. The size of the parameter samples was also increased from 100 to 600. Therefore, the total number of simulations increased from "2×2×100=400" to "6×3×600=10,800" .

4.Both reviewers commented on the equal weights (probabilities) used for the alternative scenarios and models. To address the comment, we have added a section to the revised manuscript to discuss the new sensitivity analysis results using different weight values for the scenarios and models.

5.Reviewer 2 commented on the description of the governing equations for the PAWS model. To address this comment, we have added an appendix to describe the governing equations and parameters of PAWS in detail.

General Evaluation

The topic is interesting and relevant for the scope of HESS. However, the paper does not clarify if there are new significant results achieved and the innovations are not clear. In particular hierarchical global sensitivity analysis was already implemented by Dai & Ye (2015), several applications are reported in literature (see for example Dai et al., 2017; Dai et al., 2019) and the new contribution with this paper to the technique remains unclear. The suggestion is a deeper analysis of the work and therefore it should be completely reviewed.

Response

We thank the reviewer for the positive evaluation of this manuscript and constructive comments. To provide a better physical interpretation and deeper analysis of this study, we have completely revised the uncertainty framework and uncertain inputs, as we described in the summary of the main revisions. We also implemented a new sensitivity analysis method and bin algorithm to estimate the sensitivity indices for new parameter groups. All the sensitivity analysis results and discussion of this study have been totally revised and updated accordingly.

In terms of emphasizing the innovation of this research, in the revised manuscript, we have added text to highlight the new methodology and algorithm used as follows:

"We also improved the hierarchical sensitivity analysis methodology by introducing new parameter groups into the uncertainty framework and implementing new algorithms to make the assessment of global sensitivity analysis for large-scale PBHMs computationally affordable. This study is the first to implement a comprehensive hierarchical sensitivity analysis method in relation to a complex and large-scale PBHM."

"A new set of subdivided parametric sensitivity indices was first defined to provide more detailed information for parametric sensitivities. Because of the high complexity and dimensionality of this model, the highly efficient parameter sampling method of Latin hypercube sampling (LHS) and a binning method were applied to estimate the sensitivity indices. We also investigated the effects of prior weights on the climate scenarios and numerical models."

We also revised the text to emphasize its novelty:

"By implementing the hierarchical sensitivity analysis method, we aim to provide a pilot example of comprehensive global sensitivity analysis for large-scale PBHMs considering all uncertainty sources instead of only parameters and investigate the most important source of uncertainty for modeling hydrological processes in the Amazon."

Comment 1

Page 3 Line 68 - Hierarchical global sensitivity is not implemented for the first time. New aspects are related to parameter sampling technique and the general framework is applied to PAWS+CLM hydrological model for the first time. This is not explained inside the paper.

Response

We agree with the reviewer. We have revised and added some specific descriptions of the new aspects of this study in the introduction, such as

"This study is the first to implement a comprehensive hierarchical sensitivity analysis method in relation to a complex and large-scale PBHM."

"A new set of subdivided parametric sensitivity indices was first defined to provide more detailed information for parametric sensitivities."

Comment 2

Page 4 Line 98 - Parameter $\alpha$ should have the dimension of the inverse of a length $(L-1)$ and not dimensionless. Physical meaning of the Van Genuchten parameters used for the sensitivity analysis should be reported.

Response

We thank the reviewer for pointing this out. We have revised the unit of parameter $\alpha$. More descriptions of the necessity to investigate the sensitivity of Van Genuchten parameters were also added to the introduction:

"We consider the Van Genuchten parameters $\alpha$ and N here because the correlation between $\alpha$ and N can largely affect the soil water release and infiltration processes in the vadose zone (Pan et al., 2011)."

Comment 3

Page 7 Line 175 - Prior weights for models and scenarios may affect output results. The

choice of equal weights should be motivated. An interesting point might be studying the variability of results with respect to different weights. This could be a useful tool to understand different sources of uncertainty.

Response

We thank the reviewer for pointing this out. As described in the summary of the main revisions, we have added section 3.5 to explore the influence of prior weights for the models and scenarios.

Comment 4

Page 10 Line 245 – Conductivities K have wrong measurement units.

Response

We thank the reviewer for pointing this out. We have revised the units throughout the manuscript.

Comment 5

Page 10 Line 255 – It is not clear which outputs are reported in Figure 5, if they are spatial averaged or not. Comments to Figure 5c needs a more detailed explanation (and figure reference is not 4 but 5)

Response

The original Figure 5 shows the outputs for the spatially averaged results. However, since we have completely revised the manuscript, we acquired new results and replaced this figure with Figure 4 in the revised manuscript with more detailed captions.

Comment 6

Page 11 Line 290 - An appendix with main model equations should be included for reader understanding.

Response

We thank the reviewer for the suggestion. We have added an appendix to describe the main model equations and parameters for readers to better understand the model.

Comment 7

Figure captions are too short and only acronyms of variables are reported. They should be more exhaustive.

Response

Yes, we have added more descriptions in the figure captions of the revised manuscript.

Comment 8

Formulas and indices need references.

Response

Yes, we have added references for the formulas and indices. For example, Eq. (1) and Eqs. (5)-(7).

Comment 9

Physical interpretation of results is very poor and absent in general. Substantial conclusions of the work are not highlighted. It is not clear if the hierarchical sensitivity analysis is a good tool to capture output sensitivity related to several uncertainties or not.

Response

We have completely revised the sensitivity analysis framework and results following the reviewer's comments. More physical interpretations have been added to the results since the new scenario, model, and parametric uncertain inputs have more physical meaning. The discussion and conclusion of this manuscript have also been totally revised to highlight our findings, such as

"On the basis of the results of this study, we suggest that when modelers use sophisti-
cated hydrological simulators such as PAWS, they should pay attention to the weather variable values at approximately 12:00 noon (always the daily peak values), investigate the thickness of groundwater aquifers near rivers and adjust the parameters of the vadose zone."

We have also added descriptions of the advantages of this hierarchical sensitivity analysis method, such as

"The sensitivity analysis results can provide key information on uncertainty sources for modelers and greatly improve the model calibration and uncertainty analysis processes. By categorizing multiple uncertainties into processes and placing them into a proper layer in a hierarchical framework, this advanced hierarchical sensitivity analysis method can largely reduce the computational cost associated with complex, large-scale hydrological models. Its combination with Latin hypercube sampling and the binning method can further decrease the computational cost."

References

Pan, F., Zhu, J., Ye, M., Pachepsky, Y. A., & Wu, Y. S. (2011). Sensitivity analysis of unsaturated flow and contaminant transport with correlated parameters. Journal of Hydrology (Amsterdam), 397(3-4), 238-249.

Pelletier, J. D., Broxton, P. D., Hazenberg, P., Zeng, X., Troch, P. A., & Niu, G. Y., et al. (2016). A gridded global data set of soil, intact regolith, and sedimentary deposit thicknesses for regional and global land surface modeling. Journal of Advances in Modeling Earth Systems, 8(1), 41-65.

Please also note the supplement to this comment:
https://www.hydrol-earth-syst-sci-discuss.net/hess-2019-246/hess-2019-246-AC2-supplement.pdf

246, 2019.